# GENERALIZED ANOMALY DETECTION WITH KNOWLEDGE EXPOSURE: THE DUAL EFFECTS OF AUGMENTATION

## ABSTRACT

Anomaly detection involves identifying samples that deviate from the training data. While previous methods have demonstrated significant performance, our experiments reveal that their generalization ability declines substantially when faced with slight shifts in the test data. This limitation stems from an underlying assumption: these methods generally expect the distribution of normal test samples to closely resemble that of the training set, while anomalies are presumed to be far from this distribution. However, in real-world scenarios, test samples often experience varying degrees of distributional shift while retaining their semantic consistency. The ability to generalize successfully to semantically preserved transformations while accurately detecting normal samples whose semantic meaning has changed as anomalies is critical for a model's trustworthiness and reliability. For instance, while a rotation may alter the semantic meaning of a car in the context of anomaly detection, it typically preserves the meaning of an apple. Yet, current methods, particularly those based on contrastive learning, are likely to detect both as anomalies. This complexity underscores the need for dynamic learning procedures grounded in a deeper understanding of outliers. To address this, we propose a novel approach called Knowledge Exposure (KE), which incorporates external knowledge to interpret concept dynamics and distinguish between transformations that induce semantic shifts. Our approach improves generalization by leveraging insights from a pre-trained CLIP model to assess the significance of anomalies for each concept. Evaluations on datasets such as CIFAR-10, CIFAR-100, SVHN demonstrate superior performance compared to previous methods, validating the effectiveness of our approach.

## 1 INTRODUCTION

Anomaly detection aims to identify samples that significantly deviate from the training data distribution. This task is crucial for the development of reliable machine learning systems and has various critical applications, such as marker discovery in biomedical data Schlegl et al. (2017) and video surveillance Luo et al. (2017). Unlike traditional classification approaches, which require a well-sampled outlier class, anomaly detection must function with poorly sampled or nonexistent outlier classes. This requires the use of one-class classification methods that model the distribution of inlier data to detect outliers Zimek et al. (2012); Sabokrou et al. (2018); Zaheer et al. (2022); Jewell et al. (2022).

Current test benchmarks for anomaly detection methods exhibit significant inductive biases that undermine their effectiveness in real-world settings. These benchmarks often assume that normal samples have a distribution very similar to the training set during test time, while anomalies are distributed much further away. This assumption does not hold true in practical scenarios, where real-world test samples often contain various levels of distribution shift which can maintain or change the semantic concept of the sample. This inductive bias leads to flawed testing protocols that encourage methods to have a strong bias towards a low level of diversity in normal data, which is detrimental to their real-world deployability. For instance, state-of-the-art (SOTA) methods like CSI Tack et al. (2020a) place the decision boundary extremely close to the in-distribution samples, making it easy to detect far out-of-distribution samples but ignoring variations within the inlier sets. As a

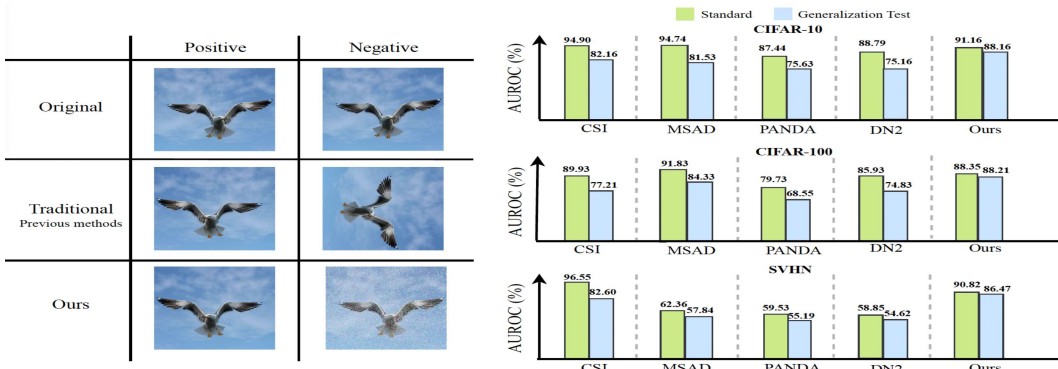

Figure 1: Comparative study on standard and our proposed anomaly detection setups (generalization test and realistic anomaly detection) using the CIFAR-10 Krizhevsky et al. (2009), CIFAR-100 Krizhevsky et al. (2009), and SVHN Netzer et al. (2018) datasets (details of the setups in Section 3). The left side shows images under traditional and our method transformations. Our method dynamically selected a transformation for each class through Knowledge Exposure (KE). In this example instead of traditionally selecting 90 degree rotation as negative pair (which will confuse model during training because rotated a rotated bird is still a bird and it can occur in the real world) we use KE to select Gaussian noise for this class. The right side presents AUROC percentages for different methods, emphasizing that while other methods suffer significant performance drops on more realistic setups due to overfitting and lack of generalizability, our method consistently maintains higher AUROC scores. This demonstrates the robustness and superior effectiveness of our approach in dynamic anomaly detection through KE.

result, under realistic conditions, these methods fail to develop reliable anomaly detection capabilities. To address this issue, future research needs to shift focus from merely improving performance on existing benchmarks to developing methods that can generalize well to more realistic conditions.

Anomaly detection methods typically use the same approach for all concepts or classes during the training. Usually, methods follow a similar pattern in learning the intrinsic idea of normalcy. For instance, in a dataset with multiple classes, samples are collected uniformly, and even in some methods, training samples are augmented before learning to enhance the learning process. While this approach is logical for balanced image classification, it may not be suitable for anomaly detection, especially with augmentation, as it could alter the normality or abnormality of samples. We aim for our model to handle different types of data augmentation while maintaining its performance and generalization. However, in anomaly detection, some forms of augmentation can change the meaning of the class and cause the samples to be labeled as abnormal. Therefore, the detector must recognize when a particular type of transformation causes the samples to be classified as anomalous while still handling others effectively.

To train an anomaly detector, creating pseudo-abnormal samples from normal ones is a common solution, as often there are no actual labeled abnormal samples during training. Different approaches are used, such as generative models, which can be computationally expensive Mirzaei et al. (2023); Pourreza et al. (2021). Another solution is the generation of pseudo-labels by transforming inlier samples through methods like rotation Tack et al. (2020a); Sohn et al. (2020). Some other methods use external datasets to expose the model to abnormal samples and train a detector Hendrycks et al. (2018). However, while these methods might work well, none accurately learn the decision boundary required to identify subtle discrepancies around boundaries. Augmentation-based methods that generate abnormal pseudo-samples are better at generating boundary samples, but selecting the transformation to generate pseudo-anomaly samples is challenging. For example, rotation doesn't necessarily change the semantics of an object. For some classes, such as apples, the original apple is considered normal, while the rotated one is considered abnormal, which is nonsensical. This leads to a huge inductive bias, where the performance of rotation detection has a linear correlation with the anomaly detection task, resulting in very low generalization. Moreover, the models are not robust to simple transformations. *This paper addresses the key question of which transformations preserve semantics and should be robust, and which should be detectable as anomalies. It also explores*

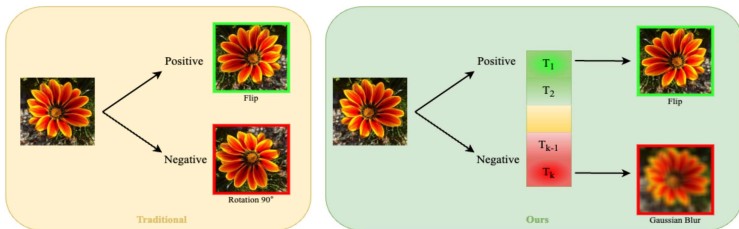

Figure 2: Showcasing the key differences between previous methods, which create pseudo-abnormal/negative samples through augmentation, and ours (right), where we dynamically select the appropriate transformations.

*how a model can automatically determine during training whether transformed samples retain their semantics, allowing for the adjustment of positive and negative pairs in conservative learning for anomaly detection.* Answering this question improves the generation of abnormal pseudo-samples, leading to better decision boundaries, enhanced generalization, and more human-like performance. As aforementioned, certain transformations, such as rotation, can have dual effects: leading to semantic shifts in some objects while not affecting others. Recent research Wang et al. (2023) demonstrates a linear correlation between rotation and semantic alterations, although the assumption of anonymity is not universally valid. Additionally, efforts are made to enhance model robustness during testing against convolutional shift variations like Gaussian noise Bai et al. (2023). Developing a robust model to handle non-semantic shifting and accurately identifying transformed semantic samples presents challenges, particularly when relying on human supervision. The key aspect is to discern whether an object is sensitive to specific transformations, termed non-transformation-agnostic, or whether it remains invariant despite changes, termed transformation-agnostic. For instance, while flowers are commonly observed in various orientations, cars rotated at 90 degrees are less frequently encountered in real-world scenarios. Thus, flowers are considered transformation-agnostic, whereas cars are non-transformation-agnostic. See Fig. 2-Left. Previous methods require generating anomaly (i.e., negative) samples to train the detector, treating rotations as negative pairs and flips as positive ones. However, for rotation-agnostic objects like flowers, rotation does not change the semantics in the context of anomaly detection. In contrast, for objects like cars, rotation can significantly alter their semantics. Our proposed method dynamically selects the most appropriate transformation for each object type. As shown in Fig. 2-Right, we automatically investigate $K$ transformations, $T_1, \ldots, T_K$, to identify the most meaningful choices for defining negative and positive pairs. The primary challenge lies in the inherent bias of the training data, which often lacks the real-world diversity necessary to capture the core features across different transformations.

To address this issue, we propose the "Knowledge Exposure" method, leveraging the capabilities of the CLIP Radford et al. (2021) model, which has been trained on a vast dataset. The CLIP's understanding of whether a concept remains invariant across different transformations serves as a valuable asset. If a concept is transformation-agnostic, the CLIP represents both the original and transformed versions similarly, owing to its exposure to a diverse range of transformations during training. Conversely, if a concept is non-transformation-agnostic, the representations differ, as the CLIP encounters fewer instances of the transformed version (see Figure 3). This insight guides our approach, which draws inspiration from reconstruction methods Xu et al. (2015); Sabokrou et al. (2016); Sakurada & Yairi (2014); Zhai et al. (2016); Zhou & Paffenroth (2017); Zong et al. (2018); Chong & Tay (2017); Gong et al. (2019); Perera et al. (2019); Abati et al. (2019); Alain & Bengio (2014); Zimmerer et al. (2018); Cong et al. (2011); Jewell et al. (2022). In essence, normal samples, which are frequently encountered, exhibit minimal loss during reconstruction. By employing this principle, we aim to identify and prioritize transformation-agnostic concepts, thereby enhancing the model's robustness against non-semantic shifting.

## 2 METHOD

**Motivation:** Contrastive learning is a common approach for anomaly detection and is widely used in the literature Tack et al. (2020a); Sohn et al. (2020). As we mentioned, these methods mostly consider rotated in-distribution (ID) samples as negative examples and other samples or, for instance,

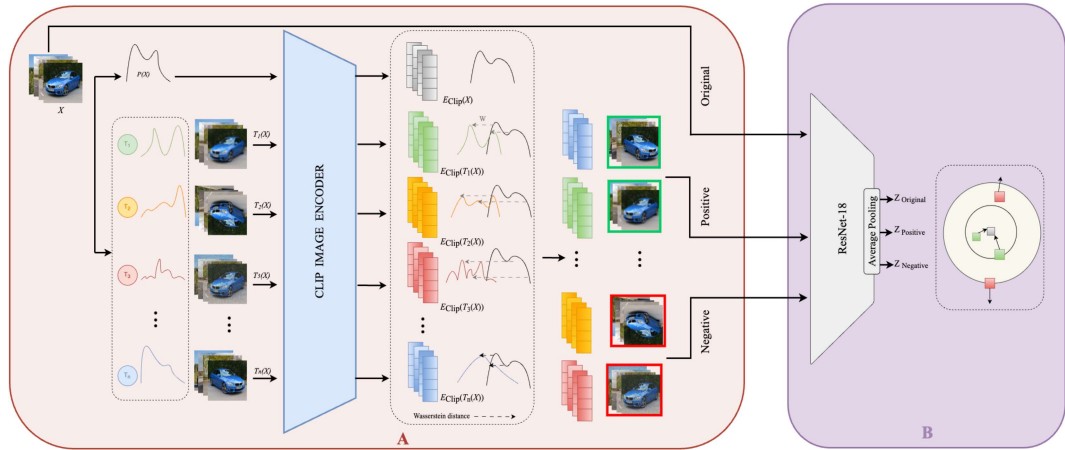

Figure 3: The training stage of our anomaly detection framework involves three key steps. (left:) Dynamic generating positive and Negative samples:First, we apply various geometric and non-geometric transformations to images of each class. Next, we feed both the original and transformed images into the CLIP image encoder, computing the Wasserstein distance between the distribution of the transformed images and the original images from encoder output. (Right:)Learning anomaly detector: Finally, we use the transformations with the smallest distance as positive pairs and those with the greatest distance as negative pairs in contrastive learning. This process ensures effective representation learning for anomaly detection.

their flipped versions as positive examples. This approach, however, performs well on standard benchmarks but significantly drops when tested for generalization. This is due to two main problems: (1) Considering a fixed augmentation for all ID samples is flawed, as some augmentations preserve the semantics while others do not, which is not suitable in the context of anomaly detection. For example, the model may mistakenly try to maximize the distance between the representation of a cat and a rotated cat. (2) Such models learn a shortcut between being an anomaly and that fixed rotation, leading to a linear correlation between rotation and anomaly detection Wang et al. (2023). To address these issues, we propose dynamically selecting the augmentation that leads to the positive and negative pairs, thereby avoiding the aforementioned problems.

**Problem Definition** Formally, let $\mathcal{X}$ be a dataset where the task is anomaly detection, and let $\mathcal{T}$ be a set of transformations. The goal is to perform supervised contrastive learning, where the positive and negative pairs of each sample are its augmented versions.

For each sample $x \in \mathcal{X}$, we denote its augmented versions under transformations $\mathcal{T}$ as $\mathcal{T}(x) = \{T(x) \mid T \in \mathcal{T}\}$, which either preserve the semantics or change them in the context of anomaly detection concepts.

Positive pairs consist of a sample $x$ and its augmentation $x^+$, where the transformation $T_+ \in \mathcal{T}$ maintains the semantics of $x$. Negative pairs consist of a sample $x$ and its augmentation $x^-$, where the transformation $T_- \in \mathcal{T}$ alters the semantics of $x$ and is contextually unsuitable for anomaly detection. The objective of the supervised contrastive learning task is to maximize the similarity between positive pairs and minimize the similarity between negative pairs, effectively learning representations that distinguish normal samples from anomalies.

The contrastive learning loss for a sample $x \in \mathcal{X}$ is defined as follows:

$$\mathcal{L}_{\text{contrastive}} = -\log \frac{\exp(\text{sim}(z, z^+)/\tau)}{\sum_{z' \in \mathcal{Z}} \exp(\text{sim}(z, z')/\tau)}$$

where $z$ is the representation of $x$, $z^+$ is the representation of the positive pair $x^+$, $\mathcal{Z}$ is the set of all representations including positive and negative pairs, $\text{sim}(\cdot, \cdot)$ denotes a similarity function (e.g., cosine similarity), $\tau$ is a temperature parameter.

This loss encourages the representations of positive pairs to be similar while ensuring that representations of negative pairs are dissimilar. Mathematically, the task is to find transformations $T_- \in \mathcal{T}$ that lead to negative pairs based on the concept, and $T_+ \in \mathcal{T}$ that lead to positive pairs. This process would adapt accordingly for a different dataset $\mathcal{X}'$. Understanding the dynamics of $\mathcal{T}$ based on $\mathcal{X}$, while many of those transformations have never been seen among inlier samples, is very challenging and can even be deemed infeasible.

**Knowledge Exposure (KE):** The samples in dataset $\mathcal{X}$ lack the diversity needed to analyze which transformations are typical and which are anomalous. In fact, most samples occur in only one position in the dataset. Understanding which augmentations or transformations preserve the conceptual essence requires a broad perspective and deep insight cultivated by exposure to a wide array of real-world scenarios. Therefore, we are motivated to leverage the CLIP image encoder, which has been trained on 400 million image and text pairs across various contexts. The rationale behind this approach is that CLIP possesses a comprehensive understanding of the real world, enabling it to discern between transformations that maintain the conceptual integrity of an image and those that significantly alter it. By leveraging the extensive training data seen by CLIP, we can identify which augmentations are encountered rarely or frequently during its training, providing valuable insights into their effects on visual concepts.

One of the main paradigms for anomaly detection is reconstruction learning, where it's claimed that learning an encoder-decoder network on dataset $\mathcal{X}$ will yield low reconstruction error for in-distribution (ID) samples and high error for out-of-distribution (OOD) samples during testing. This concept extends to the latent space, where it's observed that the latent representations of inlier samples are close to each other while those of outliers are far apart Sabokrou et al. (2018).

Inspired by these observations, we propose leveraging the CLIP image encoder to represent dataset $\mathcal{X}$ as $E_{\text{CLIP}}(\mathcal{X})$. We hypothesize that the distribution of these representations will be close to augmentations that preserve the semantic content and distant from those that alter it significantly. Consequently, we augment the dataset using all transformations $\{T_1(\mathcal{X}), \ldots, T_k(\mathcal{X})\}$ and obtain representations for each augmented dataset with $E_{\text{CLIP}}$, resulting in $\{(E_{\text{CLIP}}(\mathcal{X}), E_{\text{CLIP}}(T_1(\mathcal{X}))), \ldots, (E_{\text{CLIP}}(\mathcal{X}), E_{\text{CLIP}}(T_k(\mathcal{X})))\}$.

For a semantic-preserving transformation, we expect each sample from the original dataset to be similar to all transformed images. Measuring the similarity of the entire dataset is necessary, and the Wasserstein distance $(W)$ is ideal for this purpose. It quantifies the minimum work required to transform one distribution into another, considering differences in values and positions, and providing a comprehensive understanding of how augmentations affect the dataset. Mathematically, it is formulated as follows:

$$W(E_{\text{CLIP}}(T_i(\mathcal{X})), E_{\text{CLIP}}(T_j(\mathcal{X}))) = \inf_{\gamma \in \Gamma(E_{\text{CLIP}}(T_i(\mathcal{X})), E_{\text{CLIP}}(T_j(\mathcal{X})))} \mathbb{E}_{(x,y) \sim \gamma}[d(x,y)]$$

where $\Gamma(E_{\text{CLIP}}(T_i(\mathcal{X})), E_{\text{CLIP}}(T_j(\mathcal{X})))$ denotes the set of all possible couplings (joint distributions) of $E_{\text{CLIP}}(T_i(\mathcal{X}))$ and $E_{\text{CLIP}}(T_j(\mathcal{X}))$, and $d(x,y)$ is the distance between $x$ and $y$. Smaller W-distances indicate positive pairs (where the augmentation does not significantly change the concept from normal to abnormal), while greater distances denote negative pairs. Specifically:

$$j_+ = \arg\min_i W(E_{\text{CLIP}}(\mathcal{X}), E_{\text{CLIP}}(T_i(\mathcal{X}))) \quad \{x, x^+\} = \{x, T_{j_+}(x)\}$$

$$j_- = \arg\max_i W(E_{\text{CLIP}}(\mathcal{X}), E_{\text{CLIP}}(T_i(\mathcal{X}))) \quad \{x, x^-\} = \{x, T_{j_-}(x)\}$$

This formulation aims to identify which transformations lead to representations that closely resemble the distribution of the original dataset $(x)$ and which result in representations that deviate significantly $(x^-)$.

In this approach, to achieve better performance, we select $K$ positive samples, $\{x^+\} = T_{j_+}^{1..K}(X)$, which are the augmentations with the smallest Wasserstein distances from the original data, indicating minimal conceptual change. Similarly, we select $K$ negative samples, $\{x^-\} = T_{j_-}^{1..K}(X)$, which are the augmentations with the largest Wasserstein distances, indicating significant conceptual changes. Here, $K$ is a hyper-parameter, and its effect on performance is analyzed in section 3.

By using these multiple positive and negative samples, the model is trained to maximize the cosine similarity between the anchor $x$ and its positive samples $\{x^+\}$, while minimizing the cosine similarity between the anchor $x$ and its negative samples $\{x^-\}$ (see Figure **??**). The contrastive loss function $L_{\text{contrastive}}$ for an anchor $x$ with multiple positive and negative samples can be formulated as follows:

$$\mathcal{L}_{\text{contrastive}}(x, \{x^+\}, \{x^-\}) := -\frac{1}{|\{x^+\}|} \log \frac{\sum_{x' \in \{x^+\}} \exp(\text{sim}(z(x), z(x'))/\tau)}{\sum_{x' \in \{x^+\} \cup \{x^-\}} \exp(\text{sim}(z(x), z(x'))/\tau)} \quad (1)$$

where $z(x)$ is the representation of $x$ learned by the model (in this case, we consider ResNet-18).

One question that may arise is why we don't use the CLIP representation directly for anomaly detection. If we were to feed individual samples and their augmentations into CLIP and calculate the distance of their representations from the original dataset, the differences might not be significant enough to detect anomalies reliably. Instead, we compare these differences across the entire dataset to determine whether an augmentation is negative, applying this understanding in the contrastive loss to emphasize it. We will demonstrate this through an experiment in section 3.

**Anomaly Scoring:** To calculate anomaly scores using our method, we start by identifying positive (normal) and negative (anomalous) pairs through Knowledge Exposure. We train a ResNet-18 model in a contrastive learning paradigm using these pairs. This involves maximizing the similarity between representations of positive pairs while minimizing the similarity between representations of negative pairs. After training the ResNet-18 model, we extract feature representations for all samples in the dataset. These feature representations are then fed into a one-class Support Vector Machine (SVM), which is specifically trained to distinguish whether a new data point belongs to the same distribution as the training data (normal). The one-class SVM assigns anomaly scores to each sample based on its feature representation. Samples that fall within the learned boundary of normal samples receive lower anomaly scores, indicating they are normal. Conversely, samples that fall outside this boundary receive higher anomaly scores, indicating they are anomalies. The one-class SVM outputs score 1 for normal and -1 for anomaly samples, respectively. The details of the implementation can be found in the appendix.

## 3 Experiments and Results

In this section, we demonstrate how existing anomaly detection methods struggle under our generalization protocols, revealing that current anomaly detection benchmarks are inadequate for assessing OOD generalization in real-world scenarios. We illustrate the limitations of these methods in effectively handling semantic-preserving transformations, which are crucial for practical applications. Subsequently, we proposed the KE method, which significantly improves performance and generalization under these rigorous evaluation conditions. Our findings underscore the need for a paradigm shift in the evaluation of anomaly detection models to ensure their reliability and usability in real-world settings.

The KE mechanism effectively determined which transformations constituted positive and negative samples for contrastive learning. As depicted in Figure 4, for classes such as Bus, Train, and Motorcycle in CIFAR-100, KE identified the flip transformation as positive while recognizing the 90-degree rotation as negative due to the significant alteration of the concept in terms of anomaly detection. Similarly, for flower images, the KE mechanism deemed the 90-degree rotation as a positive transformation and color glass blur as a negative one, aligning well with the inherent semantics of these images. This dynamic selection underscores the capability of KE to discern contextually appropriate augmentations. For more visual results, you can refer to Figure 5 in the appendix.

For our experiments, we utilized three benchmark datasets: CIFAR-10, CIFAR-100 (focused on coarse-grained setting, which contains 20 superclasses), and SVHN (see the Appendix C for descriptions of these datasets and setups used for generalization evaluation). The results of these experiments will be reported by averaging the Area Under the Receiver Operating Characteristic (AUROC) across all classes. Following the one-class classification approach used in the literature, we evaluated our method in a one-vs-rest setting. We discuss implementation details, reproducibility of results, and code availability in section C. We have evaluated our methods on standard datasets (SD) and also extended standard datasets for OOD generalizationKhazaie et al. (2023). In these datasets,

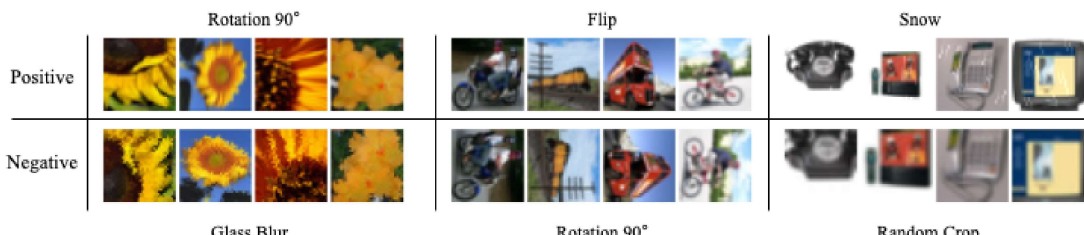

Figure 4: An illustration of the KE mechanism's ability to determine positive and negative transformations is as follows. For vehicles, KE identified the flip transformation as positive, while a 90-degree rotation was marked as negative. In contrast, for sunflowers, a 90-degree rotation was considered positive, with the glass blur identified as negative. For phones, snow noise was classified as a positive transformation, whereas random cropping was deemed negative. These choices are intuitive: a flip does not significantly alter the appearance of vehicles, but a 90-degree rotation does. Similarly, for phones, random cropping disrupts their natural appearance more than a simple rotation. Additional CIFAR-100 class images are provided in Appendix A.

the test data of standard dataset are extended by adding some augmentation samples. The aim of this experiment is for evaluating anomaly detectores are generalized against unseen OOD data or not which is crucial for real world application because in real world scenario we don't have any control over environment.

**Test Setups**: The first setup follows the Standard Detection (SD) protocol, where one class is designated as inliers, and all other classes are considered outliers(with a different distribution from training data).

However, this setup suffers from significant flaws and inductive biases. It assumes that normal samples during test time have a distribution very similar to the training set, while anomalies are distributed much further away. In real-world scenarios, test samples often exhibit various levels of distribution shifts while maintaining semantic consistency. This setup makes it easier to detect anomalies that are far from normal samples but does not account for the variations within the inlier set. Consequently, methods that simply enhance performance on existing benchmarks without addressing these nuances do not advance the field toward more practical and applicable techniques. The next setup is designed to *evaluate generalization* in the context of anomaly detection. we aim to create a benchmark that includes sufficient intra-class distribution shift on a wide range of transformations. In this setup, models trained on data distribution, are expected to be robust to image transformations. We consider images of one class and their transformations as normal samples, while images of other classes and their transformations are treated as abnormal. The reason we used transformations for images of other classes is that current methods create a shortcut between transformation and being classified as an anomaly; by applying transformations to outliers as well, we ensure that the model cannot rely on transformations alone to detect anomalies. We called this test Semantic-Preserving Detection (SPD).

Here, we focus on presenting the results from the SD and SPD setups, as they are the main focus of our evaluation. These setups provide a more stringent and realistic assessment of anomaly detection methods. Tables 1, 2 and 5 summarize the performance of various anomaly detection methods such as CSITack et al. (2020a), MSADReiss & Hoshen (2023), PANDAReiss et al. (2021), DN2Bergman et al. (2020) and our proposed method, on CIFAR-10, CIFAR-100, and SVHN datasets on all setups. The results are reported in terms of AUROC, averaged across all classes. Our method demonstrates significant improvements over existing methods across all datasets and SPD setup and is comparable to others in term of SD. These results highlight the effectiveness of our method under challenging and realistic evaluation conditions, confirming the benefits of dynamic augmentation selection and using contrastive learning framework in enhancing anomaly detection capabilities. While our method

Table 1: AUROC (%) of various novelty detection methods trained on one-class setting of CIFAR-10 dataset.

| Classes | CSI | | MSAD | | PANDA | | DN2 | | Ours | |
|---|---|---|---|---|---|---|---|---|---|---|
| | SD | SPD | SD | SPD | SD | SPD | SD | SPD | SD | SPD |
| Plane | 90.0 | 79.02 | 95.87 | 79.34 | 88.61 | 72.46 | 87.91 | 77.56 | 91.36 | 93.86 |
| Car | 99.13 | 89.53 | 97.90 | 91.42 | 95.69 | 87.58 | 96.15 | 79.42 | 91.56 | 89.22 |
| Bird | 94.46 | 80.27 | 88.90 | 68.76 | 74.61 | 62.22 | 79.43 | 63.38 | 92.46 | 92.08 |
| Cat | 87.42 | 72.99 | 88.35 | 69.43 | 80.0 | 69.15 | 78.22 | 64.88 | 91.32 | 83.21 |
| Deer | 95.48 | 80.39 | 94.34 | 80.67 | 87.66 | 75.45 | 91.13 | 79.60 | 91.03 | 88.58 |
| Dog | 93.23 | 78.03 | 94.86 | 80.68 | 84.61 | 71.28 | 86.03 | 67.74 | 90.22 | 89.88 |
| Frog | 96.27 | 83.01 | 96.60 | 86.49 | 89.58 | 79.94 | 90.43 | 79.45 | 92.38 | 92.61 |
| Horse | 98.94 | 87.79 | 96.51 | 84.60 | 89.17 | 73.01 | 89.74 | 72.38 | 91.59 | 91.36 |
| Ship | 97.90 | 86.25 | 96.68 | 82.39 | 90.33 | 78.63 | 93.89 | 82.66 | 92.21 | 88.70 |
| Truck | 96.26 | 84.42 | 97.40 | 91.56 | 94.15 | 86.66 | 94.99 | 84.57 | 87.51 | 76.62 |
| Mean | 94.90 | 82.16 | **94.74** | 81.53 | 87.44 | 75.63 | 88.79 | 75.16 | 91.16 | **88.16** |

Table 2: AUROC (%) of various novelty detection methods trained on one-class setting of SVHN dataset.

| Classes | CSI | | MSAD | | PANDA | | DN2 | | Ours | |
|---|---|---|---|---|---|---|---|---|---|---|
| | SD | SPD | SD | SPD | SD | SPD | SD | SPD | SD | SPD |
| Digit 0 | 96.85 | 82.49 | 68.16 | 60.80 | 62.16 | 54.95 | 65.14 | 56.25 | 94.29 | 89.97 |
| Digit 1 | 90.99 | 80.06 | 64.89 | 59.17 | 66.28 | 58.29 | 67.82 | 62.20 | 89.12 | 90.41 |
| Digit 2 | 97.43 | 84.26 | 64.48 | 58.80 | 60.35 | 56.55 | 60.31 | 55.73 | 90.32 | 86.01 |
| Digit 3 | 94.69 | 79.75 | 60.49 | 57.19 | 58.66 | 55.86 | 57.12 | 54.41 | 92.65 | 70.01 |
| Digit 4 | 98.78 | 83.96 | 65.49 | 58.18 | 63.69 | 57.95 | 65.24 | 59.43 | 88.47 | 91.94 |
| Digit 5 | 97.00 | 83.41 | 60.23 | 58.05 | 57.81 | 55.39 | 60.93 | 55.04 | 88.58 | 79.12 |
| Digit 6 | 96.82 | 84.56 | 58.11 | 55.76 | 55.70 | 53.06 | 51.78 | 49.87 | 88.90 | 90.61 |
| Digit 7 | 98.70 | 82.13 | 57.42 | 53.41 | 60.54 | 55.75 | 60.96 | 56.90 | 93.78 | 93.81 |
| Digit 8 | 96.71 | 81.68 | 65.26 | 60.18 | 57.19 | 54.00 | 51.27 | 49.69 | 90.66 | 85.10 |
| Digit 9 | 97.53 | 83.80 | 59.23 | 56.93 | 52.99 | 50.14 | 47.91 | 46.78 | 91.51 | 87.82 |
| Mean | **96.55** | 82.60 | 62.36 | 57.84 | 59.53 | 55.19 | 58.85 | 54.62 | 90.82 | **86.47** |

provides better generalization, it achieves comparable performance to existing methods on SD setup which reveals that those methods achieves slightly higher performance through sacrificing generalization and subcutaneous overfitting on the training datasets distribution without considering generalization which is more essential for real world applications.

## 3.1 ABLIATION STUDY

**The Effect of K Pairs for Contrastive Learning:** In our evaluation with exploring different hyperparameters we explored the impact of using multiple positive and negative pairs in the contrastive learning framework, referred to as the K-Pairs approach. Using KE approach will empower us to select transformations which leads the K-Pairs approach enhances the model's ability to more meaningful

Table 3: Using only the raw features from different pre-trained backbones for anomaly detection on the CIFAR-100 dataset. The backbones that start with C are all CLIP image encoders.

| Backbone | AUROC |
|---|---|
| C_RN50 | 82.18 |
| C_RN101 | 83.65 |
| C_RN50x4 | 82.62 |
| C_RN50x16 | 80.25 |
| C_RN50x64 | 78.94 |
| C_ViT-B/16 | 76.82 |
| C_ViT-B/32 | 75.95 |
| C_ViT-L/14 | 81.56 |
| ViT-B/16 | 59.07 |
| ConvNeXt/B | 62.14 |
| CSWin/B | 86.42 |
| PatchConvnet/B60 | 63.93 |
| VAN/L | 66.49 |
| PVT/B4 | 70.69 |
| **Ours(ResNet-18)** | 88.35 |

Table 4: The Effect of Different Values of the **K** Hyperparameter on Model Generalization on CIFAR-10 dataset. The results show that using **K=2** enhances the model's generalization by exposing it to two distinct transformations as positive and negative data.

| Classes | K=1 | K=2 |
|---|---|---|
| Class 0 | 75.26 | 81.18 |
| Class 1 | 82.12 | 82.47 |
| Class 2 | 79.11 | 84.33 |
| Class 3 | 79.97 | 81.61 |
| Class 4 | 88.35 | 82.26 |
| Class 5 | 80.78 | 80.68 |
| Class 6 | 79.27 | 84.43 |
| Class 7 | 83.43 | 86.89 |
| Class 8 | 77.85 | 82.20 |
| Class 9 | 63.03 | 79.41 |
| Mean | 78.91 | 82.54 |

representation by increasing the diversity of both positive and negative pairs, yields to robustness of model. Specificaly, this apporach enables model to focus on relative transformations that are more representive of real data distributions. Rather than relying on single positive, the model benefits from a broader range of transformation. At the same time using multiple negative pairs of the same data (selected from KE) ensures that the learned representation are distinct and seperated, reducing the risk of features collaps. As it is represented in table 4 the result of using K-Pairs approach leads to imporved generalization and robustness compare to using single positive negative approach, providing the model with stronger foundation for anomaly detection generalization.

**Raw Features of CLIP for Anomaly Detection:** We conducted an ablation experiment to assess the performance of using the raw features of the CLIP image encoder directly for anomaly detection. Specifically, we applied a one-class SVM on these raw features to detect anomalies under the SSA and SPA setups. For better comparison, we also included the raw features of other image encoders. The results indicated that using raw CLIP features did not lead to robust performance, yielding significantly lower AUROCs compared to our proposed method. Table 3 demonstrates that raw features from the CLIP encoder are not sufficient for effective anomaly detection under our rigorous testing protocols. Our ResNet-18 model, despite having significantly fewer parameters, outperformed all pre-trained networks with many more parameters. This superior performance can be attributed to the ResNet-18 being trained through Knowledge Exposure and further learning the nuances with contrastive learning, which enabled it to achieve better results than the raw features from all pre-trained networks. As mentioned previously, large-scale encoders are generally robust against most transformations.

**Alternative Encoders for Knowledge Exposure:** To evaluate the flexibility of our framework, we experimented with different image encoders instead of the CLIP image encoder. We used DINOv2 Oquab et al. (2023); Darcet et al. (2023) and pure Wide-ResNet-50 Zagoruyko & Komodakis (2016) as alternatives encoder for knowledge exposure. The results, evaluated through human supervision, indicated that resnet will not generate sufficient encoding to distinguish transformations from each other and DINOv2 did not generalize well across all augmentations and transformations, leading to lower performance compared to CLIP. However, our core idea is not restricted to the CLIP model. The framework remains effective with different encoders, although CLIP yielded the best results due to its extensive training on a diverse dataset. The performance was assessed based on the distance obtained by knowledge exposure from both models.

**The Effect of Transformations on Anomaly Class**: To specifically address the impact of certain transformations on anomaly detection models, we designed an ablation experiment. We aimed to evaluate how these transformations, when applied to anomaly classes, influence the models' ability to correctly classify anomalies. We observed that the models exhibited a bias, often resulting in false negatives, where anomalies were misclassified as normal due to the presence of similarly transformed normal samples. As illustrated in Figure 6, the distribution of distances extracted from the DN2 model for the "Car" class of the CIFAR-10 dataset shows that our assumption of the model forming a strong relation to transformations holds. This figure demonstrates that rotated instances partially overlap with the distribution of normal instances.

## 4 CONCLUSION AND LIMITATIONS

In this paper, we address an important shortcoming in the literature on anomaly detection. We demonstrate that each transformation or augmentation has two facets: one that alters the meaning of samples and one that does not, particularly in the context of anomaly detection. Previous methods have overlooked this aspect, resulting in significant limitations in real-world applications and a notable lack of generalization. To understand the dual nature of each augmentation, we leverage the knowledge from a pre-trained model and develop a contrastive learning-based method. This method dynamically selects negative and positive pairs, taking into account both facets of each augmentation. Our results indicate that the proposed method exhibits greater generalization and performs significantly better in realistic scenarios. However, a limitation of our approach is its reliance on a pre-trained network to provide the necessary knowledge for all transformations. This dependency may restrict the method's applicability in situations where such pre-trained models are not available or suitable.

