# REFERENCES

Davide Abati, Angelo Porrello, Simone Calderara, and Rita Cucchiara. Latent space autoregression for novelty detection. In *Proceedings of the IEEE/CVF Conference on Computer Vision and Pattern Recognition*, pp. 481–490, 2019.

Guillaume Alain and Yoshua Bengio. What regularized auto-encoders learn from the data-generating distribution. *The Journal of Machine Learning Research*, 15(1):3563–3593, 2014.

Haoyue Bai, Gregory Canal, Xuefeng Du, Jeongyeol Kwon, Robert D Nowak, and Yixuan Li. Feed two birds with one scone: Exploiting wild data for both out-of-distribution generalization and detection. In *International Conference on Machine Learning*, pp. 1454–1471. PMLR, 2023.

Liron Bergman, Niv Cohen, and Yedid Hoshen. Deep nearest neighbor anomaly detection. *arXiv preprint arXiv:2002.10445*, 2020.

Ting Chen, Simon Kornblith, Mohammad Norouzi, and Geoffrey Hinton. A simple framework for contrastive learning of visual representations. In *International conference on machine learning*, pp. 1597–1607. PMLR, 2020.

Xuhai Chen, Yue Han, and Jiangning Zhang. A zero-/few-shot anomaly classification and segmentation method for cvpr 2023 vand workshop challenge tracks 1&2: 1st place on zero-shot ad and 4th place on few-shot ad. *arXiv preprint arXiv:2305.17382*, 2023.

Yong Shean Chong and Yong Haur Tay. Abnormal event detection in videos using spatiotemporal autoencoder. In *International symposium on neural networks*, pp. 189–196. Springer, 2017.

Yang Cong, Junsong Yuan, and Ji Liu. Sparse reconstruction cost for abnormal event detection. In *CVPR 2011*, pp. 3449–3456. IEEE, 2011.

Timothée Darcet, Maxime Oquab, Julien Mairal, and Piotr Bojanowski. Vision transformers need registers. In *The Twelfth International Conference on Learning Representations*, 2023.

Sepideh Esmaeilpour, Bing Liu, Eric Robertson, and Lei Shu. Zero-shot out-of-distribution detection based on the pre-trained model clip. In *Proceedings of the AAAI conference on artificial intelligence*, volume 36, pp. 6568–6576, 2022.

William Falcon and Kyunghyun Cho. A framework for contrastive self-supervised learning and designing a new approach. *arXiv preprint arXiv:2009.00104*, 2020.

Dong Gong, Lingqiao Liu, Vuong Le, Budhaditya Saha, Moussa Reda Mansour, Svetha Venkatesh, and Anton van den Hengel. Memorizing normality to detect anomaly: Memory-augmented deep autoencoder for unsupervised anomaly detection. In *Proceedings of the IEEE/CVF International Conference on Computer Vision*, pp. 1705–1714, 2019.

Kaiming He, Haoqi Fan, Yuxin Wu, Saining Xie, and Ross Girshick. Momentum contrast for unsupervised visual representation learning. In *Proceedings of the IEEE/CVF conference on computer vision and pattern recognition*, pp. 9729–9738, 2020.

Dan Hendrycks, Mantas Mazeika, and Thomas Dietterich. Deep anomaly detection with outlier exposure. *arXiv preprint arXiv:1812.04606*, 2018.

Jongheon Jeong, Yang Zou, Taewan Kim, Dongqing Zhang, Avinash Ravichandran, and Onkar Dabeer. Winclip: Zero-/few-shot anomaly classification and segmentation. In *Proceedings of the IEEE/CVF Conference on Computer Vision and Pattern Recognition*, pp. 19606–19616, 2023.

John Taylor Jewell, Vahid Reza Khazaie, and Yalda Mohsenzadeh. One-class learned encoder-decoder network with adversarial context masking for novelty detection. In *Proceedings of the IEEE/CVF Winter Conference on Applications of Computer Vision*, pp. 3591–3601, 2022.

Vahid Reza Khazaie, Anthony Wong, and Mohammad Sabokrou. Towards realistic out-of-distribution detection: A novel evaluation framework for improving generalization in ood detection. *arXiv preprint arXiv:2211.10892*, 2023.

Alex Krizhevsky, Geoffrey Hinton, et al. Learning multiple layers of features from tiny images. *Technical Report*, 2009.

Aodong Li, Chen Qiu, Marius Kloft, Padhraic Smyth, Maja Rudolph, and Stephan Mandt. Zero-shot anomaly detection via batch normalization. *Advances in Neural Information Processing Systems*, 36, 2024.

Philipp Liznerski, Lukas Ruff, Robert A Vandermeulen, Billy Joe Franks, Klaus-Robert Müller, and Marius Kloft. Exposing outlier exposure: What can be learned from few, one, and zero outlier images. *arXiv preprint arXiv:2205.11474*, 2022.

Weixin Luo, Wen Liu, and Shenghua Gao. A revisit of sparse coding based anomaly detection in stacked rnn framework. In *Proceedings of the IEEE International Conference on Computer Vision*, pp. 341–349, 2017.

Hossein Mirzaei, Mohammadreza Salehi, Sajjad Shahabi, Efstratios Gavves, Cees GM Snoek, Mohammad Sabokrou, and Mohammad Hossein Rohban. Fake it till you make it: Near-distribution novelty detection by score-based generative models. *ICLR*, 2023.

Yuval Netzer, Tao Wang, Adam Coates, Alessandro Bissacco, Bo Wu, and A Ng. The street view house numbers (svhn) dataset. *Technical Report*, 2018.

Maxime Oquab, Timothée Darcet, Théo Moutakanni, Huy V Vo, Marc Szafraniec, Vasil Khalidov, Pierre Fernandez, Daniel HAZIZA, Francisco Massa, Alaaeldin El-Nouby, et al. Dinov2: Learning robust visual features without supervision. *Transactions on Machine Learning Research*, 2023.

Pramuditha Perera, Ramesh Nallapati, and Bing Xiang. Ocgan: One-class novelty detection using gans with constrained latent representations. In *Proceedings of the IEEE/CVF Conference on Computer Vision and Pattern Recognition*, pp. 2898–2906, 2019.

Masoud Pourreza, Bahram Mohammadi, Mostafa Khaki, Samir Bouindour, Hichem Snoussi, and Mohammad Sabokrou. G2d: Generate to detect anomaly. In *Proceedings of the IEEE/CVF Winter Conference on Applications of Computer Vision*, pp. 2003–2012, 2021.

Alec Radford, Jong Wook Kim, Chris Hallacy, Aditya Ramesh, Gabriel Goh, Sandhini Agarwal, Girish Sastry, Amanda Askell, Pamela Mishkin, Jack Clark, et al. Learning transferable visual models from natural language supervision. In *International conference on machine learning*, pp. 8748–8763. PMLR, 2021.

Tal Reiss and Yedid Hoshen. Mean-shifted contrastive loss for anomaly detection. In *Proceedings of the AAAI Conference on Artificial Intelligence*, volume 37, pp. 2155–2162, 2023.

Tal Reiss, Niv Cohen, Liron Bergman, and Yedid Hoshen. Panda: Adapting pretrained features for anomaly detection and segmentation. In *Proceedings of the IEEE/CVF Conference on Computer Vision and Pattern Recognition*, pp. 2806–2814, 2021.

Mohammad Sabokrou, Mahmood Fathy, and Mojtaba Hoseini. Video anomaly detection and localisation based on the sparsity and reconstruction error of auto-encoder. *Electronics Letters*, 52 (13):1122–1124, 2016.

Mohammad Sabokrou, Mohammad Khalooei, Mahmood Fathy, and Ehsan Adeli. Adversarially learned one-class classifier for novelty detection. In *Proceedings of the IEEE Conference on Computer Vision and Pattern Recognition*, pp. 3379–3388, 2018.

Mayu Sakurada and Takehisa Yairi. Anomaly detection using autoencoders with nonlinear dimensionality reduction. In *Proceedings of the MLSDA 2014 2nd Workshop on Machine Learning for Sensory Data Analysis*, pp. 4–11, 2014.

Thomas Schlegl, Philipp Seeböck, Sebastian M Waldstein, Ursula Schmidt-Erfurth, and Georg Langs. Unsupervised anomaly detection with generative adversarial networks to guide marker discovery. In *International conference on information processing in medical imaging*, pp. 146–157. Springer, 2017.

Florian Schroff, Dmitry Kalenichenko, and James Philbin. Facenet: A unified embedding for face recognition and clustering. In *Proceedings of the IEEE conference on computer vision and pattern recognition*, pp. 815–823, 2015.

Kihyuk Sohn, Chun-Liang Li, Jinsung Yoon, Minho Jin, and Tomas Pfister. Learning and evaluating representations for deep one-class classification. In *International Conference on Learning Representations*, 2020.

Jihoon Tack, Sangwoo Mo, Jongheon Jeong, and Jinwoo Shin. Csi: Novelty detection via contrastive learning on distributionally shifted instances. In H. Larochelle, M. Ranzato, R. Hadsell, M.F. Balcan, and H. Lin (eds.), *Advances in Neural Information Processing Systems*, volume 33, pp. 11839–11852. Curran Associates, Inc., 2020a. URL `https://proceedings.neurips.cc/paper_files/paper/2020/file/8965f76632d7672e7d3cf29c87ecaa0c-Paper.pdf`.

Jihoon Tack, Sangwoo Mo, Jongheon Jeong, and Jinwoo Shin. Csi: Novelty detection via contrastive learning on distributionally shifted instances. *Advances in neural information processing systems*, 33:11839–11852, 2020b.

Guodong Wang, Yunhong Wang, Xiuguo Bao, and Di Huang. Rotation has two sides: Evaluating data augmentation for deep one-class classification. In *The Twelfth International Conference on Learning Representations*, 2023.

Dan Xu, Elisa Ricci, Yan Yan, Jingkuan Song, and Nicu Sebe. Learning deep representations of appearance and motion for anomalous event detection. *arXiv preprint arXiv:1510.01553*, 2015.

Sergey Zagoruyko and Nikos Komodakis. Wide residual networks. *European Conference on Computer Vision (ECCV)*, 2016.

M. Zaigham Zaheer, Arif Mahmood, M. Haris Khan, Mattia Segu, Fisher Yu, and Seung-Ik Lee. Generative cooperative learning for unsupervised video anomaly detection. In *Proceedings of the IEEE/CVF Conference on Computer Vision and Pattern Recognition (CVPR)*, pp. 14744–14754, June 2022.

Shuangfei Zhai, Yu Cheng, Weining Lu, and Zhongfei Zhang. Deep structured energy based models for anomaly detection. In *International Conference on Machine Learning*, pp. 1100–1109. PMLR, 2016.

Chong Zhou and Randy C Paffenroth. Anomaly detection with robust deep autoencoders. In *Proceedings of the 23rd ACM SIGKDD international conference on knowledge discovery and data mining*, pp. 665–674, 2017.

Qihang Zhou, Guansong Pang, Yu Tian, Shibo He, and Jiming Chen. Anomalyclip: Object-agnostic prompt learning for zero-shot anomaly detection. *arXiv preprint arXiv:2310.18961*, 2023.

Arthur Zimek, Erich Schubert, and Hans-Peter Kriegel. A survey on unsupervised outlier detection in high-dimensional numerical data. *Statistical Analysis and Data Mining: The ASA Data Science Journal*, 5(5):363–387, 2012.

David Zimmerer, Simon AA Kohl, Jens Petersen, Fabian Isensee, and Klaus H Maier-Hein. Context-encoding variational autoencoder for unsupervised anomaly detection. *arXiv preprint arXiv:1812.05941*, 2018.

Bo Zong, Qi Song, Martin Renqiang Min, Wei Cheng, Cristian Lumezanu, Daeki Cho, and Haifeng Chen. Deep autoencoding gaussian mixture model for unsupervised anomaly detection. In *International Conference on Learning Representations*, 2018.

APPENDIX

# A   ADDITIONAL EXPERIMENTS AND RESULTS

Figure 5: Visualization of dynamically chosen negative and positive pairs for each class through Knowledge Exposure (KE) from randomly selected classes in CIFAR-100.

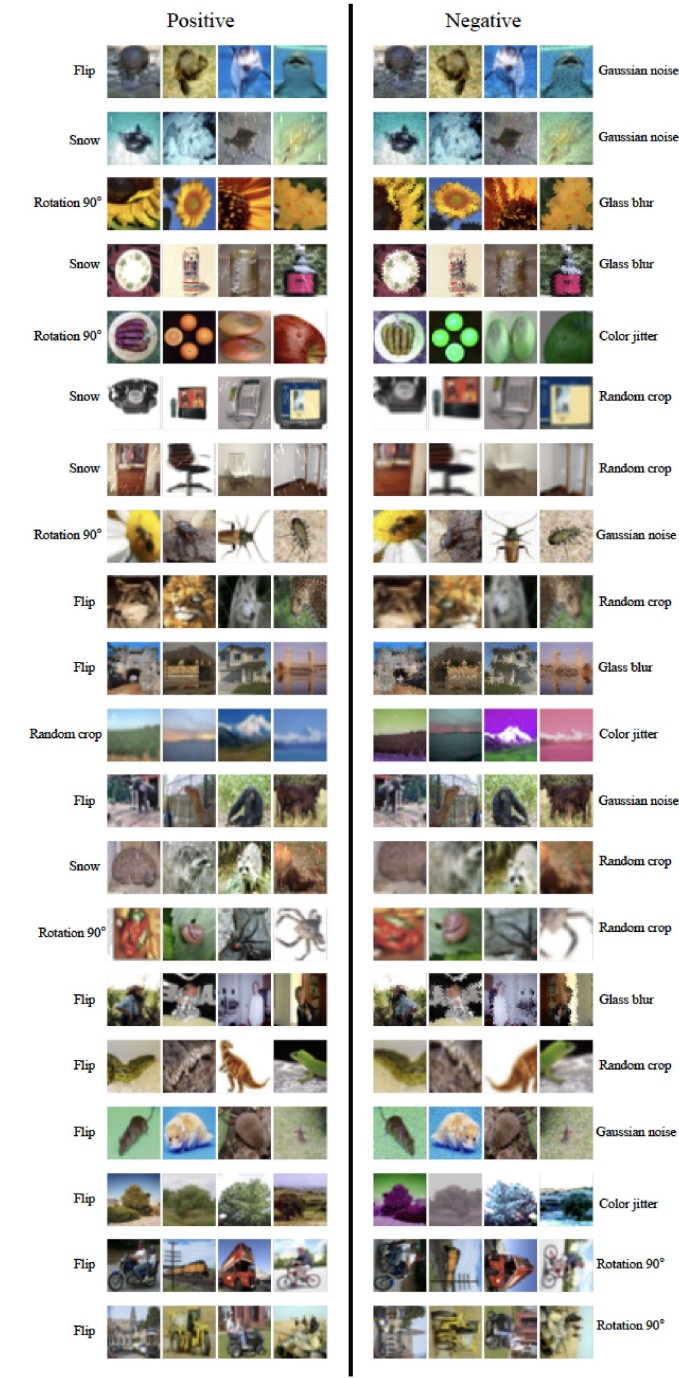

The effectiveness of Knowledge Exposure (KE) in dynamically selecting appropriate negative and positive pairs is evident. For instance, the identification of color jitter as a negative pair for fruits, sky scenes, or other color-dependent concepts is logical, as these transformations significantly disrupt the

color integrity crucial for anomaly detection. Similarly, for objects such as cars, buses, and bicycles, a 90-degree rotation is correctly identified as a negative pair since such rotations would render these objects anomalous in real-world scenarios. Conversely, for objects like apples or oranges, a 90-degree rotation is sensibly selected as a positive pair, as this transformation does not alter their conceptual integrity. Figure 5 clearly illustrates that KE performs effectively across various cases, reinforcing the results presented in the paper and highlighting the robustness and accuracy of our anomaly detection framework.

Table 5: AUROC (%) of various novelty detection methods trained on one-class setting of CIFAR-100 dataset.

| Classes | CSI | | MSAD | | PANDA | | DN2 | | Ours | |
|---|---|---|---|---|---|---|---|---|---|---|
| | SD | SPD | SD | SPD | SD | SPD | SD | SPD | SD | SPD |
| Class 0 | 87.15 | 74.57 | 92.15 | 96.09 | 76.40 | 66.78 | 81.71 | 73.17 | 88.27 | 88.28 |
| Class 1 | 84.92 | 72.53 | 87.41 | 84.49 | 80.48 | 72.18 | 81.99 | 75.55 | 88.49 | 89.12 |
| Class 2 | 88.95 | 82.71 | 95.86 | 82.05 | 89.44 | 77.09 | 92.91 | 82.91 | 85.88 | 91.84 |
| Class 3 | 84.61 | 74.77 | 91.04 | 82.65 | 68.98 | 58.96 | 82.82 | 64.73 | 88.93 | 85.18 |
| Class 4 | 93.01 | 77.35 | 95.49 | 85.36 | 85.80 | 69.70 | 91.64 | 70.10 | 90.14 | 90.00 |
| Class 5 | 82.57 | 71.66 | 92.50 | 92.92 | 77.61 | 63.99 | 89.92 | 68.52 | 90.49 | 90.26 |
| Class 6 | 92.63 | 78.94 | 92.63 | 82.55 | 78.08 | 70.27 | 88.78 | 73.12 | 88.54 | 89.63 |
| Class 7 | 83.90 | 73.48 | 88.78 | 83.14 | 77.16 | 61.74 | 82.84 | 67.21 | 87.18 | 89.81 |
| Class 8 | 93.37 | 70.57 | 95.26 | 90.46 | 81.24 | 66.98 | 89.21 | 70.26 | 92.69 | 82.27 |
| Class 9 | 95.54 | 87.26 | 91.33 | 88.26 | 80.69 | 84.73 | 87.39 | 87.69 | 78.37 | 92.19 |
| Class 10 | 93.59 | 84.12 | 94.32 | 74.97 | 89.01 | 73.25 | 93.86 | 78.74 | 89.11 | 67.50 |
| Class 11 | 90.03 | 83.94 | 89.91 | 82.03 | 69.88 | 66.38 | 78.19 | 77.95 | 89.17 | 94.19 |
| Class 12 | 91.27 | 78.48 | 90.67 | 79.54 | 78.44 | 68.25 | 80.74 | 81.86 | 83.15 | 84.86 |
| Class 13 | 82.32 | 72.88 | 84.26 | 84.30 | 75.48 | 59.15 | 77.00 | 67.94 | 88.84 | 89.12 |
| Class 14 | 94.90 | 81.28 | 94.22 | 81.34 | 84.47 | 71.84 | 88.88 | 74.91 | 90.82 | 90.97 |
| Class 15 | 85.76 | 69.03 | 86.77 | 83.34 | 69.46 | 61.03 | 80.88 | 77.87 | 89.74 | 92.64 |
| Class 16 | 84.13 | 77.12 | 85.28 | 84.45 | 74.00 | 70.75 | 76.39 | 74.30 | 90.89 | 86.19 |
| Class 17 | 97.26 | 73.55 | 97.00 | 80.84 | 94.26 | 63.16 | 96.05 | 70.62 | 86.65 | 86.37 |
| Class 18 | 96.82 | 77.19 | 97.02 | 84.83 | 84.52 | 71.83 | 88.77 | 79.25 | 95.77 | 97.61 |
| Class 19 | 95.99 | 82.80 | 94.79 | 83.08 | 79.22 | 72.95 | 88.72 | 79.91 | 84.04 | 86.18 |
| Mean | 89.93 | 77.21 | **91.83** | 84.33 | 79.73 | 68.55 | 85.93 | 74.83 | 88.35 | **88.21** |

# B  RELATED WORKS

One of the most widely used techniques in anomaly detection involves employing self-supervised methods, which generate pseudo-labels from the data itself. Contrastive learning Falcon & Cho (2020), a self-supervised technique, has shown remarkable success in visual representation learning He et al. (2020); Chen et al. (2020). This method focuses on learning representations by contrasting positive and negative samples, ensuring that representations of similar (positive) instances are closer together, while representations of dissimilar (negative) instances are further apart. Common loss functions used include triplet loss, NT-Xent loss, and InfoNCE Schroff et al. (2015); Chen et al. (2020). Recent research has indicated that transformations once thought to be harmful in traditional contrastive learning can be beneficial in out-of-distribution (OOD) detection. In the Contrasting Shifted Instances (CSI) method Tack et al. (2020a), in addition to contrasting a given sample

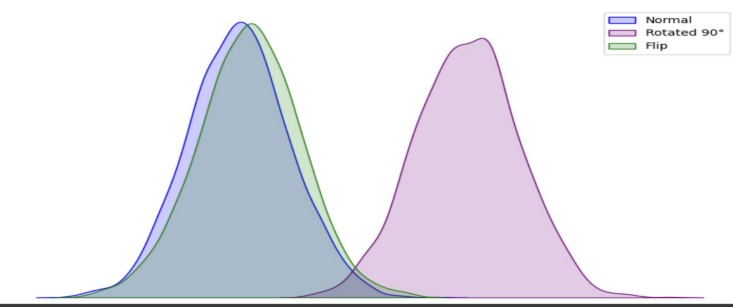

Figure 6: This figure illustrates the distribution of distances under different transformations: the blue distribution represents the original (normal) images, the green distribution represents flipped images, and the red distribution represents 90-degree rotated images of all instances from the "car" class in the CIFAR-10 dataset. The significant overlap between the rotated and normal data indicates that the model has a strong relation to rotation.

with other instances, the training scheme also contrasts the sample against distributionally shifted augmentations of itself, which enhances OOD detection Tack et al. (2020b).Transformations can result in either semantics-preserving or semantics-shifting images, depending on the class, which should be considered when selecting positive and negative pairs. Wang et al. (2023) addressed this issue in rotation transformation, but this principle can also apply to several geometric and shifting transformations.

Additionally, several methods use knowledge from pre-trained models to identify normal data patterns. These methods, including DN2Bergman et al. (2020), PANDAReiss et al. (2021), and MSADReiss & Hoshen (2023), first employ the pre-trained models to extract features that represent typical data points. Then, they utilize techniques like k-nearest neighbors (KNN) and Gaussian mixture models (GMM) to assess the distance of new data points from the established set of normal features. This distance is used to calculate an anomaly score.

Furthermore, outlier exposure (OE) is a new technique proposed for anomaly detection tasks, which uses an auxiliary dataset of outliers Hendrycks et al. (2018). The main problem with OE is its reliance on a large and diverse outlier dataset during training, which may not be readily available in many practical scenarios. Additionally, the learned representations may not generalize well to unseen outlier distributions, and in the case of irrelevant outliers, performance decreases.

Recently, large pre-trained vision-language models, trained using millions of image-text pairs Radford et al. (2021), have demonstrated strong zero-shot recognition ability in various vision tasks, including anomaly detection. These models rely on the ability to transfer knowledge from auxiliary data to identify unseen anomalies. Early approaches, such as CLIP-AD Liznerski et al. (2022), ZOC Esmaeilpour et al. (2022), and ACR Li et al. (2024), which require tuning on auxiliary data for each target dataset, have been proposed for anomaly classification. Recent approaches focus on both anomaly segmentation and classification. For effective anomaly segmentation, WinCLIP Jeong et al. (2023) employs a wide range of hand-designed text prompts and multiple forward passes of image patches. To improve the modeling of local visual semantics, VAND Chen et al. (2023) introduces learnable linear projection techniques. However, these methods encounter issues because text prompt embeddings lack sufficient generalization, leading to reduced accuracy in identifying anomalies associated with diverse, unseen object semantics. AnomalyCLIP Zhou et al. (2023) tackles these challenges by adapting to diverse datasets after being trained on a general dataset. It uses only two trainable, object-independent text prompts for identifying anomalies and segments images with a single forward pass.

However, these methods rely on learning text prompts to capture anomalies, which may not generalize as well as directly learning from image data. Additionally, they use a combination of global and local loss functions, which could be less efficient compared to our adaptive contrastive learning on image features.

## C    REPRODUCIBILITY AND IMPLEMENTATION DETAILS

### C.1    DATASET DETAILS

We conducted experiments using the CIFAR-10, CIFAR-100, and SVHN data sets for anomaly detection. The CIFAR-10 dataset comprises 60,000 color images of 32x32 size, evenly distributed across 10 classes, with each class containing 6,000 images. The CIFAR-100 dataset extends to 100 classes, each with 600 images. Both CIFAR datasets are split into training sets of 50,000 images and testing sets of 10,000 images. The SVHN dataset, derived from Google Street View, consists of more than 600,000 32x32 color images of house numbers, used for digit recognition and classification.

### C.2    PREPROCESSING PROCEDURE

To create the real-world dataset, we used Knowledge Exposure (KE) to generate a list of distances for each original class from various augmentations. These distances were sorted to identify negative augmentations as anomalies. Additionally, in SSA, alongside KE, we employed human supervision to refine the dataset and applied other augmentations and transformations not considered negative in the training process. For example, using KE, Gaussian noise and glass blur were identified as the most negative augmentations for cars. However, in the real world, rotation is considered an anomaly for cars. Therefore, in creating the real-world dataset, we used human supervision to label rotated cars as anomalies.

To create a comprehensive list of anomalies for each class in each dataset, we initially implemented the aforementioned protocols using KE, supplemented by human supervision for accurate and real-world scenario selection. We then loaded the original dataset and iterated through it, applying a randomly selected transformation from our predefined list to each data point. We evaluated whether this transformation was classified as an anomaly based on the previously assembled anomaly transformation list.

For preprocessing, the start with augmenting, the original dataset using and then feed them into the CLIP image encoder to produce the representations. Then use extracted representations in Wasserstein algorithm for scoring each augmentaion.

### C.3    TRAINING PROCEDURE

For training, we utilized ResNet-18 without a classification head as the architecture for our neural network. We trained our model for 50 epochs using the SGD optimizer with a learning rate of 0.01, momentum of 0.9, and a weight decay of 5e-4 for each experiment. We used AUROC as the anomaly detection metric, where the AUROC value ranges from 0 to 1, with values closer to 1 indicating better classifier performance. For loss function we utilize the InfoNCE loss Chen et al. (2020) with 0.2 temperature value for better convergence rate. Our findings indicate that using 2 for **K** hyperprameter enhances the generalization which can be interpreted as model being exposed to two distinct augmentations or transformations as positive and negative data which results in constriction of model into more significant features, thereby reducing the likelihood of overfitting to irrelevant features which available in Table 4. Experiments were conducted on NVIDIA RTX 3090 GPUs, and runtimes are approximately 12 hours for training on CIFAR-10.

### C.4    EVALUATION PROCEDURE

In each method discussed previously, we employed standard data and specific configurations unique to each method during their respective training phases. For instance, in the case of CSI, we preserved its fundamental augmentation. We implemented two protocols for every dataset: the standard (SD) and the generalization test (SPD). During the training phase, all models requiring training data were provided with a chosen dataset, undergoing its standard pre-processing without modifications or augmentations to the training procedure. In the testing phase, apart from augmenting the test set, no modifications were made to the test configurations of the methods. For generating augmented datasets we used the Khazaie et al. (2023) framework to be as generalized as possible in training datasets we use hyper-parameter severity equal to 1 but in the proposed dataset we use 5 for severity to become a near real world worst case scenario. For evaluation, we utilized our trained model to

extract features from training data for training a one-class SVM with sigmoid kernel. Then for any input data, we use the same model to extract the features and then predict the label, whether input is normal or anomalous.

## C.5 CODE AVAILABILITY

The code for the implementation of the model and the reproduction of the results of our experiments can be found at `https://anonymous.4open.science/r/AKKKS-FBB4/`.

.