# OpenReview forum: "Generalized Anomaly Detection with Knowledge Exposure:The Dual Effects of Augmentation"
_ICLR.cc/2025/Conference — ICLR 2025 Conference Withdrawn Submission_

### Official Review · Reviewer_6B8Z · 2024-10-26

**Soundness:** 2
**Presentation:** 1
**Contribution:** 2
**Rating:** 3
**Confidence:** 4

**Summary:**

The paper tackles th task of distribution shift at inference time for AD systems. It does so by proposing augmentations that preserve semantics while improving domain generalization. It uses CLIP as its supervision for determining if an augmentation changes the semantics . It determines the augmntations per the ntir distribution rather than per-sample using a set distance based criterion (Wasserstein). It then introduces an additional contrastive loss term that learns features invariant to semantics presrving augmentations. The xpriments show this procedure improves the invariance to the augmentations.

**Strengths:**

The task this paper tackles is doubtlessly important.We should encourage research in this topic.
The observation that CLIP cannot reliably classify instance level augmentation invariance is interesting, although the solution has limitations..

**Weaknesses:**

The paper has multiple limitations that lead the reviewer to recommend rejection:

*Augmentation semantic change procedure is not well motivated.* The paper proposes to use WD on CLIP feature distribution to check if an augmentation changes the semantics of the normal data. There are multiple flaw with this idea. First, normal data often contain multiple classes, each invariant to other augmentations. This procedure is not suitable for this (more common) case. Second, WD is expensive to compute. The fact that CLIP distance is not accurate enough on a single sample, does not mean that simple running CLIP for each sample before and after the augmentation, computing the distance for each instanc and taking the average across the set will not work. Third, the list of possible domain changs is large (infinite in fact), and contains many possible changes in noise, resolution, rotation etc. Exhaustively, enumerating all ths factors is expensive or impossible and designing augmentations for some of thse factors of variation is also impossible.

*Choice of anomaly detector and unsound experiments.* It is well established that pre-trained features extractors work better on Cifar10/100 than contrastive methods (e.g., CSI). This paper choose to us a contrastive method. However, it only compared to pretrained methods on small resnets where they do not perform well. However, they work much better (at least in the standard, no domain shift) for larger networks (e.g., ResNet152). The comparisons therefore do not show the full picture. Furthermore, they can easily incorporate augmted sampled in their training dataset, and thus will presumably perform well in the domain shifted setting. The domain shifted setting is somewhat self serving as the same procedure is used to select the augmentations for the proposed method and choosing the "Real world" test set.

*Toy datasets.* The experiments are all on toy datasets (cifar10/100, SVHN) and in the easiest setting (one vs all)

*Writing.* The paper is not well written. no citep, there are missing references, the bibliography is not in the main paper (it's in the appendix), the figures are badly scaled and not attractive.

**Questions:**

Can the authors please rerun the Cifar10/100 experiments on large pretrained encoders e.g., DinoV2 and incorporate augmentations in the training set?
Can the authors compare augmentation selection with WD and with the average pre/post augmentation CLIP distance?

---

### Official Review · Reviewer_n4HX · 2024-10-28

**Soundness:** 2
**Presentation:** 2
**Contribution:** 1
**Rating:** 3
**Confidence:** 4

**Summary:**

This paper introduces a novel method for selecting augmentations to enhance representation learning for anomaly detection by leveraging the CLIP model. The authors' experiments demonstrate that the learned representations improve the generalizability of anomaly detection, particularly in scenarios where the test distribution deviates from the training distribution.

**Strengths:**

- The paper tackles a critical challenge in anomaly detection: the generalizability of existing methods under distribution shifts at test time.
- A key limitation of many state-of-the-art methods is their dependence on hand-crafted augmentations to simulate anomalies. By leveraging CLIP to automatically select augmentations in a data-driven manner, the authors offer a compelling solution to this issue.

**Weaknesses:**

- The manuscript is poorly written, containing numerous typos, low-quality figures, and broken references.
- The paper lacks sufficient details about the proposed method. For example, the choice of the Wasserstein distance over simpler alternatives like cosine or Euclidean distance is not well justified. Additionally, the specific augmentations used in the method are not described.
- The method underperforms compared to baselines in standard evaluation settings. While the authors show improvements in a new setting, they provide an unclear description, making it difficult to assess whether the gains are simply due to adapting the method to this new evaluation.

**Questions:**

- In line 53, the authors claim that methods like CSI *place the decision boundary extremely close to the in-distribution samples*, which they argue leads to ignoring variations within the inlier set. However, they neither provide evidence to support this claim nor address it in their own method, as one-class SVMs typically create a tight decision boundary around the inliers.
- There is a broken reference in line 272.
- The authors use only simple, low-resolution datasets. The evaluation would be stronger with more challenging datasets like ImageNet30 or datasets that naturally exhibit domain shifts in the test set.
- What are the SSA and SPA setups mentioned in lines 467 and 468?
- The results for the **Alternative Encoders for Knowledge Exposure** experiment are missing. Additionally, the authors should explain why their method fails with other pretrained models, as conceptually, this should not make a significant difference.
- Reference [1] also explores anomaly detection under domain shifts and should be included in the related work.

[1] Carvalho, João, et al. "Invariant anomaly detection under distribution shifts: a causal perspective." Advances in Neural Information Processing Systems 36 (2024).

---

### Official Review · Reviewer_E4VQ · 2024-11-01

**Soundness:** 2
**Presentation:** 1
**Contribution:** 1
**Rating:** 1
**Confidence:** 4

**Summary:**

Dear Authors,

Thank you for submitting to ICLR.

This paper proposes a new approach to improve the generation of pseudo-anomalous samples to support the training of anomaly detection models. The proposed method addresses practical challenges faced by conventional generation techniques, which often fail to preserve semantic information and, consequently, struggle to generalize well in scenarios involving distribution shifts in new input instances.

On the algorithmic side, the approach utilizes a pretrained CLIP model for contrastive learning to capture invariant information from the input. For the anomaly detection component, the method leverages ResNet-18 to extract image features, followed by an SVM to assign anomaly scores.

**Strengths:**

The proposed method appears interesting. Leveraging a pretrained model like CLIP to capture invariant information could be effective in certain cases. However, the approach primarily relies on a straightforward application of CLIP, and it is unclear if there is a substantial contribution beyond this baseline utilization.

**Weaknesses:**

Technical aspects:
1. Problem setup is questionable. It not very clear about what "realistic" means in thie paper. CIFAR dataset may not be sufficient to show the distribution shift, and it cannot represent many realistic scenarios. So the overall evaluation is not very convinving.
2. Using pseudo labels or data are not the only solution in real world. It is also common to use semi-supervised learning with solely normal data. The related works are not discussed is this paper.
3. Very limited contribution. This paper is a straightforward application of trained CLIP, without any explanation about why the CLIP is helpful and what is the alternative method. The anomalous score assignment method is also very simple with SVM.
4. Even empirically, the gain is not very clear from Table 1 and Table 2.

Writing:
This is poorly written paper. Here I only list few significant issues:
1. Reference are all gone; (therefore, I recommand a rejection)
2. Appendix A is mentioned but not enclosed;
3. Page 6, missing figure reference;
4. Not explanation to "CLIP";
5. Very long paragraph between page 2 and 3, which is hard to understand.

**Questions:**

Please refer to the weaknesses. While the idea of using CLIP is interesting, its specific impact on the proposed method is not clearly demonstrated. Additionally, the evaluation lacks comprehensiveness, limiting the assessment of the approach's effectiveness. Furthermore, the writing quality is poor, with significant formatting issues that make the paper hard to read.

**Details Of Ethics Concerns:**

There is no ethics concerns

---

### Official Review · Reviewer_HjKA · 2024-11-06

**Soundness:** 1
**Presentation:** 2
**Contribution:** 2
**Rating:** 3
**Confidence:** 4

**Summary:**

The paper looks into the image augmentation method used in training anomaly detectors. Specific object classes favor different image augmentation methods. The semantic meanings of some objects are preserved under transformations; but some are not. The paper proposes to carefully select semantic-preserved transformations per class to improve the generalization of anomaly detectors under covariate shifts. To do this, they use CLIP as an external knowledge base to help select transformations for an image class. The authors experimented with the proposed method using contrastive-learning based anomaly detectors as backbone models. The proposed method is shown effective in covariate shift settings.

**Strengths:**

- The paper clearly describes the misalignment augmentation problem.
- The proposed method is effective.

**Weaknesses:**

The main weakness is the untested alternatives:
- Since the proposed method already uses the pretrained CLIP as an external knowledge base, why not directly compare CLIP-based anomaly detection method (Liznerski, Philipp, et al. Exposing outlier exposure: What can be learned from few, one, and zero outlier images, TMLR)? Answering this question will increase the significance of the paper.

Other weaknesses include:
- The paper should discuss potential extensions or alternatives for non-image data types, or to clarify the scope and limitations of their method's applicability.
- More comprehensive datasets can be employed to strengthen the empirical study (Hendrycks, D. and Dietterich, T., 2019. Benchmarking neural network robustness to common corruptions and perturbations. ICLR 2019). That dataset has realistic corruptions, e.g., JPEG compression artifacts.
- The method’s performance is not that strong on standard detection protocol. Compared to CSI, there is still a large gap.

**Questions:**

Here are some comments and suggestions:
- I feel attributing the generalization issue to augmentation needs more explanation. Augmentation also helps generalization to covariate shifts, as can be seen from the performance of CSI baseline, which performs well in covariate shifts in Table 1 and 2. Maybe compare your method against CSI on different augmentation strategies to show advantages.
- Generalization problems are also studied by zero-shot methods, e.g., Li et al, Zero-shot anomaly detection via batch normalization, NeurIPS 2023. The paper should discuss that sequence of works as well in similarities, differences, complementary aspects, etc.
- Can the proposed method be a plug-in for other contrastive-learning based methods like CSI, MSDA and show improvements on them?
- Figure resolution can be improved.
- L272 missing figure reference

---

### Note · Authors · 2024-11-15

I have read and agree with the venue's withdrawal policy on behalf of myself and my co-authors.